# The Impact of TSC-1 and -2 Mutations on Response to Therapy in Malignant PEComa: A Multicenter Retrospective Analysis

**DOI:** 10.3390/genes13111932

**Published:** 2022-10-24

**Authors:** Lawrence Liu, Carina Dehner, Nikhil Grandhi, Yang Lyu, Dana C. Borcherding, John S. A. Chrisinger, Xiao Zhang, Jingqin Luo, Yu Tao, Amanda Parkes, Nam Q. Bui, Elizabeth J. Davis, Mohammed M. Milhem, Varun Monga, Mia Weiss, Brian Van Tine, Angela C. Hirbe

**Affiliations:** 1City of Hope Comprehensive Cancer Center, Duarte, CA 91010, USA; 2Department of Pathology and Immunology, Washington University School of Medicine, St. Louis, MO 63110, USA; 3Department of Medicine, Washington University School of Medicine, St. Louis, MO 63110, USA; 4Siteman Cancer Center, Washington University School of Medicine, St. Louis, MO 63110, USA; 5Carbone Cancer Center, University of Wisconsin, Madison, WI 53706, USA; 6Stanford Cancer Institute, Stanford University, Palo Alto, CA 94305, USA; 7Ingram Cancer Center, Vanderbilt University, Nashville, TN 37232, USA; 8Holden Comprehensive Cancer Center, University of Iowa, Iowa City, IA 52722, USA

**Keywords:** PEComa, angiomyolipoma, lymphangiomyomatosis, mTOR inhibitor, everolimus, sirolimus, temsirolimus

## Abstract

Background: Perivascular epithelioid cell neoplasms (PEComas) are a diverse family of mesenchymal tumors with myomelanocytic differentiation that disproportionately affect women and can be associated with tuberous sclerosis (TS). Although mTOR inhibition is widely used as first-line treatment, it is unclear what genomic alterations exist in these tumors and how they influence the response to therapy. Methods: This was a multicenter study conducted at five sites within the US. The data were collected from 1 January 2004 to 31 January 2021. We conducted a retrospective analysis to identify PEComa patients with next-generation sequencing (NGS) data and compared outcomes based on mutations. Results: No significant differences in survival were identified between *TSC-1* and *TSC-2* mutated PEComa or *TSC-1*/*-2* versus other mutations. No significant difference was seen in progression-free survival (PFS) after first-line therapy between mTOR inhibition versus other systemic therapies. Conclusions: We were unable to detect differences in survival based on genomic alterations or PFS between mTOR inhibition versus other systemic therapies. Future studies should seek to identify other drivers of *TSC-1/-2* silencing that could predict response to mTOR inhibition.

## 1. Introduction

Perivascular epithelioid cell neoplasms (PEComas) are mesenchymal tumors with myomelanocytic differentiation. This family of neoplasms consists of renal angiomyolipoma (RAL), lymphangiomyomatosis (LAM), clear cell “sugar” tumors, hepatic angiomyolipoma, PEComa not otherwise specified (PEComa-NOS), and others [1,2,3,4,5]. There are only 40–80 cases per year in the United States. In the literature, median overall survival (mOS) is typically 60 months [5]. Women are disproportionately affected by PEComas (70%), with a median age of presentation of 46 years and a range from 15–100 years [6]. While some PEComa cases are associated with tuberous sclerosis (TS), 80% of cases are sporadic. RALs occur in 80% of patients with tuberous sclerosis, while LAM occurs in 30% of women with TS [1,4,7,8]. These tumors can appear almost anywhere anatomically, and there is not a common symptom or sign associated with this neoplasm. For RALs, roughly a quarter of patients may experience flank pain, and 5% experience hematuria, but the majority will be asymptomatic and detected incidentally by imaging [1,2,4,9]. Histologically, their appearance can vary from having epithelioid features to spindle cells, and some tumors have even been found to express melanin [1,2,3,4]. The diagnosis is made by observation of combined smooth muscle and melanocytic differentiation, supported by positive staining for Melan A, HMB-45, MITF, and SMA. A subset is positive for TFE3. Occasionally, these tumors will show positivity for S100 protein, caldesmon, desmin, estrogen receptor, and progesterone receptor [1,3,4].

The majority of PEComas have a loss of function (LOF) of *TSC-1* or *-2* genes (commonly *TSC-2* LOF), leading to increased mTOR pathway activation. Therefore, TS patients are most predisposed to PEComa family tumors. A minority of tumors harbor rearrangements of the *TFE3 (Xp11)* gene locus, which leads to downstream activation of mTOR and other cancer promoting pathways [1,3,4,9,10,11,12,13].

Management involves surgery, radiation, and/or systemic therapy, depending on the tumor type, location, invasion, and spread [4,7,10,12,13,14] Given that the vast majority of PEComas are driven by mTOR hyper-activation, the widely agreed upon systemic therapy is mTOR inhibition via everolimus, sirolimus, or temsirolimus [7]. Bissler et al. performed a randomized double-blind placebo-controlled trial evaluating everolimus in patients with RAL associated with TS or sporadic LAM [7,8]. Significant activity was observed, with a 54% overall response rate (ORR) at roughly 29 months [7]. Follow-up of these patients found that for those who had a response to everolimus, tumor growth was observed after cessation of therapy [8]. Additionally, randomized controlled trials (RCTs) utilizing sirolimus in RAL and LAM showed very similar results and ORRs (40–50%) [15,16,17]. These studies of everolimus and sirolimus along with numerous case reports demonstrate a very good safety profile of mTOR inhibition with relatively low rates of adverse events [4,7,9,10,11,13,15]. The recent AMPECT trial investigated the efficacy and safety profile of ABI-009, albumin-bound sirolimus, in the treatment of malignant PEComa and demonstrated an ORR of 39%, median progression-free survival (mPFS) of 10.6 months, and a mOS of 40.6 months [13]. A study by Sanfilippo et al. reported the efficacy of various systemic therapies against PEComas: gemcitabine (ORR 20%, with mPFS of 3.4 months), anthracyclines (ORR 13%, with mPFS of 3.2 months), and antiangiogenic agents (ORR 8.3%, with mPFS of 5.4 months); mTOR inhibition was still shown to be the most effective (ORR of 41%, with a mPFS of 9 months) [14]. However, a large genomic analysis of these tumors has not been performed. Thus, it is unclear what other genomic alterations may exist in these tumors and how they influence the response to first-line mTOR inhibition.

## 2. Materials and Methods

We conducted a retrospective analysis to identify PEComa patients with next-generation sequencing (NGS) data and compared outcomes based on mutations. This was a multicenter study conducted at five sites within the US: Washington University in St. Louis, the University of Wisconsin, the University of Iowa, Stanford University, and Vanderbilt University. The data was collected from surgical pathology reports from 1 January 2004 to 31 January 2021. Forty-nine patients with surgical pathology reports and chart data were identified across the participating sites.

Data from pathology reports were collected and stored in the RedCap database. Malignant PEComa was defined as pathology demonstrating two or more of the following: primary tumor > 5 cm, infiltrative, high nuclear grade and cellularity, mitotic rate ≥ 1/50 high power field, necrosis, vascular invasion [18]. In addition to collecting data on malignant criteria, we also collected data on mutational data from NGS, surgery, radiation therapy, systemic therapy, response, metastatic disease, time to progression after each treatment line, and survival time. A flow chart demonstrating patient accrual is included in the Appendix A. A table with pathology information of each patient was included in the Appendix A.

Progression-free survival (PFS) was defined as the date of diagnosis to date of progression or death or last imaging scan. Overall survival (OS) was defined as the date of diagnosis to date of death or last follow-up. PFS and OS were thresholded at 12 and 50 months, respectively. ORR was defined as partial response (PR) or complete response (CR). Clinical benefit rate (CBR) was defined as stable disease, PR, or CR. GraphPad Prism 9.1.1 software sourced from Washington University in St. Louis, MO, US was used for statistical analyses. Empirical survival probability was estimated via Kaplan–Meier method and differences estimated via log-rank test.

## 3. Results

The patient population was primarily female (72%), and tumor type was mostly PEComas other than RAL (68%). Primary tumor location was largely uterine (24%) or retroperitoneal/kidney (26%). The majority of patients were treated with surgical resection initially (94%), and 32% experienced local recurrence (Table 1**)**. For *TSC-1* or *TSC-2* mutated PEComa treated with mTOR inhibition as first-line therapy, the ORR was 25% and CBR was 75% compared to 33% and 66%, respectively, for PEComa without the *TSC-1* or *-2* mutations also treated with mTOR inhibition (Table 2). There was a non-statistically significant trend towards improved OS in *TSC-2* mutated PEComa vs. *TSC-1* (Figure 1). Patients with *TSC-1* or *-2* mutated tumors had an initial trend towards improved OS compared other mutations, which disappeared at 50 months (Figure 2). Patients treated with first-line mTOR inhibition (regardless of mutational status) versus other systemic therapy had an initial trend towards improved PFS, which disappeared at 12 months (Figure 3).

## 4. Discussion

Although there are only 40–80 cases of PEComa per year, they are associated with a median OS of 60 months [1,2,3,4,5]. Additionally, 80% of TS patients develop RALs, and while most are benign, there is a risk of malignant transformation. Thus far, upfront mTOR inhibition has provided the best results, with the AMPECT trial recently reporting an ORR of 39%, a mPFS of 10.6 months, and a mOS of 40.6 months [13]. However, it is unclear what other genomic alterations may exist in these tumors and how they influence the response to first-line mTOR inhibition.

We sought to identify any differences in survival and progression between *TSC-1* or *-2* mutation versus other mutations. First, we demonstrated that there was no statistically significant difference in survival based on *TSC-1* versus *TSC-2* mutational status. There were only two patients with *TSC-1* mutations, which limits our ability to see a difference in response rates between *TSC-1* and *TSC-2*. Second, there was no significant difference in survival for *TSC-1* or *TSC-2* compared to other mutations. Our ORR for first-line mTOR inhibition (25% in *TSC-1* or *TSC-2* and 33.3% in other mutations) was lower in comparison to that of the AMPECT trial (39%) [13]. This suggests that there is a benefit to nab-sirolimus over the oral mTOR inhibitors, potentially due to the fact that the oral drugs can have variable absorption and incomplete target suppression. There were six patients with *TSC-1* and *TSC-2* mutations that were treated with other systemic therapies in the first-line setting. It is possible that exclusion of these patients led to worse ORR. Third, we did not find a significant difference in PFS1 with mTORi versus other therapy. However, most patients received mTOR inhibition and there were too few patients who received chemotherapy or immunotherapy first line for us to make any statistically significant conclusions.

The limitations of our study include the retrospective nature and the small sample size. Additionally, whole-exome sequencing was utilized as opposed to whole-genome sequencing, so intronic splice mutations or copy number changes would not have been detected. Future studies should seek to identify other drivers or mechanisms of *TSC-1/-2* silencing that could predict response to therapy. Additionally, the identification of TFE3 translocations may be crucial as predictors of the theoretical ineffectiveness of mTOR inhibition [19]. Additionally, novel mutations could be drivers of this rare cancer, as demonstrated in this presentation of cutaneous PEComa [20]. Identification of such mutations could provide targets for future therapies. Our understanding of the behavior and biology of rare cancers such as PEComa is possible with collaborations within sarcoma centers.

## Figures and Tables

**Figure 1 genes-13-01932-f001:**
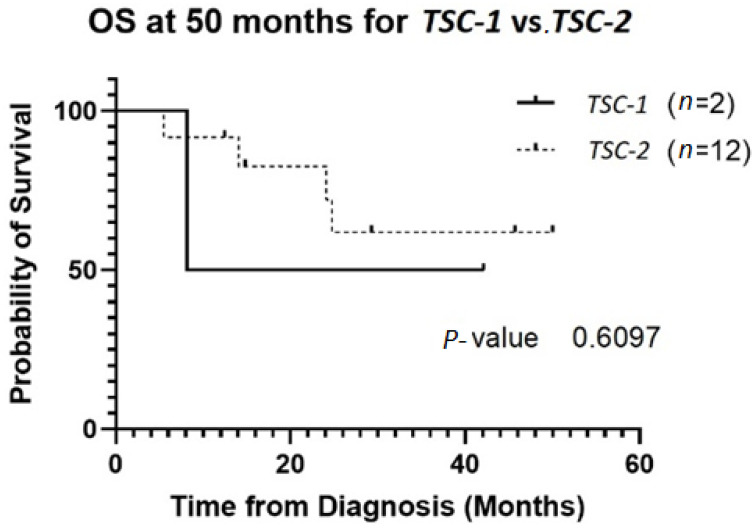
OS of patients with *TSC-1* versus *TSC-2* mutated tumors. Two patients had *TSC-1* mutations and 12 patients had *TSC-2* mutations. Four of the *TSC-2* mutated tumors were treated with local resection without systemic therapy without recurrence. Only one of the *TSC-1* mutated tumors received mTOR inhibition. Abbreviations: OS, overall survival; TSC, tuberous sclerosis complex.

**Figure 2 genes-13-01932-f002:**
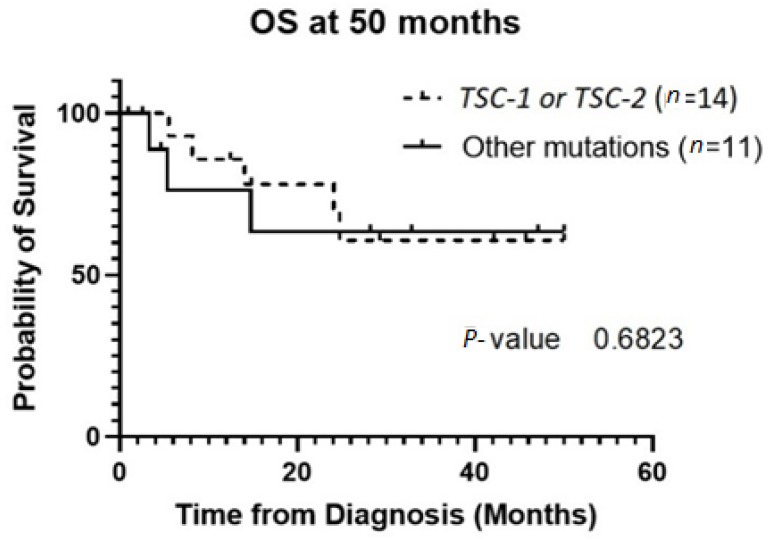
OS of the group of patients with *TSC-1* and *-2* mutated tumors versus other mutations. Patients who did not have genetic testing sent out were excluded from this analysis. Abbreviations: OS, overall survival; TSC, tuberous sclerosis complex.

**Figure 3 genes-13-01932-f003:**
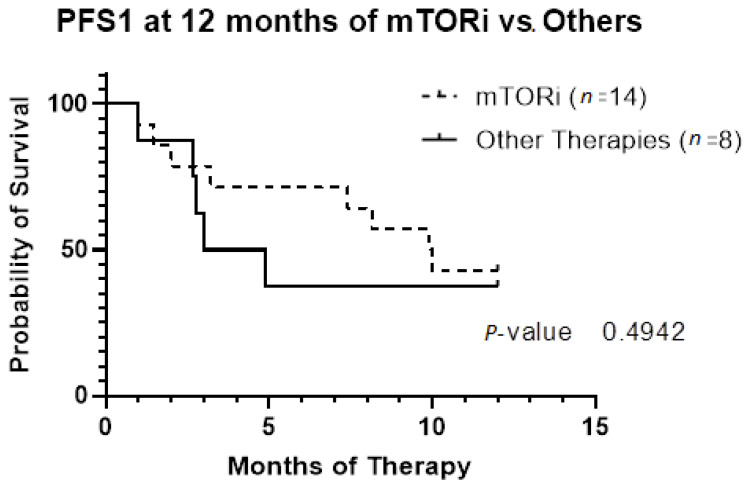
Progression-free survival based on first-line systemic therapy censored at 12 months. Patients included in this analysis underwent systemic therapy for their PEComa (n = 22). Fourteen patients had first-line therapy with mTOR inhibition. Only two patients did not meet criteria for malignant PEComa; they had initial resection with local recurrence requiring systemic therapy. Abbreviations: PFS1, progression-free survival of first-line therapy; mTORi, mTOR inhibition.

**Table 1 genes-13-01932-t001:** Baseline characteristics. * Some cases had multiple mutations.

Patient Characteristics	*N* = 49 (%)
**Sex**	
Male	14 (28)
Female	36 (72)
**Age at Diagnosis (years)**	54 (31–78) *
**Tumor Subtype**	
Angiomyolipoma	11 (22)
PEComa	34 (68)
**Primary Tumor Location**	
GI Tract	9 (18)
Soft Tissue	5 (10)
Liver	1 (2)
Kidney/Retroperitoneum	13 (26)
Uterus	12 (24)
Unknown	9 (20)
**Metastases at Presentation**	5 (10)
**Local Recurrence**	16 (32)
**Grade**	
High	18 (36)
Low	1 (2)
Malignant PEComa	27 (55)
**Surgical Resection**	47 (94)
**Treated with Systemic Therapy**	22 (44)
**Mutational Status Known ***	20 (40)
*TSC-1*	2 (4)
*TSC-2*	12 (24)
*TFE-3* Translocation	9 (18)
*TP53* Mutation	9 (18)
*RB1* Mutation	3 (6)

**Table 2 genes-13-01932-t002:** Overall response rate and clinical benefit rate.

Mutation	First-Line Therapy	*N* = 18	ORR (%)	CBR (%)
*TSC-1 or -2*	mTORi	8	25	75
	Anthracycline	0		
	Other	1 *	100	100
Other	mTORi	6	33.3	66.7
	Anthracycline	0		
	Other	3 **	0	66.7

This table compares the mutational status and response to different therapies. The column “*N*” is the number of patients who had the mutation and underwent the first-line therapy mentioned. Abbreviations: ORR, overall response rate; CBR, clinical benefit response; mTORi, mTOR inhibition. * The patient was treated with pembrolizumab and had a partial response to therapy with time to progression of 20.22 months. ** Two patients were treated with taxanes, and one was treated with arimidex since the tumor was positive for estrogen receptor.

## Data Availability

Not applicable.

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
