# Peer review of "The Impact of TSC-1 and -2 Mutations on Response to Therapy in Malignant PEComa: A Multicenter Retrospective Analysis"

_genes, 2022, doi:10.3390/genes13111932_

Round 1

Reviewer 1 Report

Dear Academic Editor and dear authors,

thank you to give me the opportunity to review this interesting paper. In this manuscript, the authors describe a retrospective study on malignant PEComa recorded in 5 different centres. 

Introduction is well written and complete. I suggest to the authors to add some reference very recent that can improve the novelty of this topic.

Cazzato G, Colagrande A, Lospalluti L, Pacello L, Lettini T, Arezzo F, Loizzi V, Lupo C, Casatta N, Cormio G, Maiorano E, Ingravallo G, Resta L. Primitive Cutaneous (P)erivascular (E)pithelioid (C)ell Tumour (PEComa): A New Case Report of a Rare Cutaneous Tumor. Genes (Basel). 2022 Jun 26;13(7):1153. doi: 10.3390/genes13071153. PMID: 35885936; PMCID: PMC9317609.

Material and methods, results and discussion are fine.

Author Response

Reviewer 1:

Dear Academic Editor and dear authors,

thank you to give me the opportunity to review this interesting paper. In this manuscript, the authors describe a retrospective study on malignant PEComa recorded in 5 different centres. 

Introduction is well written and complete. I suggest to the authors to add some reference very recent that can improve the novelty of this topic.

Cazzato G, Colagrande A, Lospalluti L, Pacello L, Lettini T, Arezzo F, Loizzi V, Lupo C, Casatta N, Cormio G, Maiorano E, Ingravallo G, Resta L. Primitive Cutaneous (P)erivascular (E)pithelioid (C)ell Tumour (PEComa): A New Case Report of a Rare Cutaneous Tumor. Genes (Basel). 2022 Jun 26;13(7):1153. doi: 10.3390/genes13071153. PMID: 35885936; PMCID: PMC9317609.

Thank you for this suggestion. We included it into our discussion section in line 168.

Material and methods, results and discussion are fine.

Thank you.

Reviewer 2 Report

This article is concise and answers the question; the main limitation, as you have pointed out, is the small number of patients selected. So it's complicated to draw a conclusion. A complementary molecular biology analysis could have allowed us to answer the question more accurately by increasing the cohort of patients. Moreover, a flow chart is missing to fully understand the selection of patients.

The introduction talks mainly about RAL and LAM but these are not the usual malignant PEComas. It would have been interesting to develop more on these PEComas-NOS ( not otherwise specified) which are usually treated with mTOR inhibitors but often less subject to TSC1/2 mutation. In front of this very rare tumor, it would also be interesting to know if a double anatomopathological reading was performed by an expert

INTRODUCTION

35 : there are also hepatic angiomyolipoma

36  : PEComas of the lung are usually LAM or clear cell “sugar” tumor of lung, I don't understand the sub-type

37 : Median survival was calculated on patients in the literature and not on the US population

38 : 70% in the literature

Materials and Methods

86 : RedCap database involves centralized review of PEComas cases by a qualified pathologist ?

RESULTS 

Add a flow chart ?

Figure 1 : little interest due to the low number of TSC1 mutated patients

Figure 2 : 11 other mutations,  excluding tumors with multiple mutations ?

An other figure to see OS in mTOR-treated patients who are TSC mutated and non-mutated ?

Table 1 : angiomylipoma are classical triphasic or épithelioid angiomyolipoma ?

What is high or low grade ? It depend on Folpe's criteria ?

Why surgical resection end local recurrence are in "Grade" ?

Table 2 : how we can have CBR without ORR for the population without mutation with other treatments ?

DISCUSSION

Role of TFE3 translocation in the literature ? Response to mTOR in this group of PEComas ?

References

there are two references 1.

Author Response

Reviewer 2:

This article is concise and answers the question; the main limitation, as you have pointed out, is the small number of patients selected. So it's complicated to draw a conclusion. A complementary molecular biology analysis could have allowed us to answer the question more accurately by increasing the cohort of patients. Moreover, a flow chart is missing to fully understand the selection of patients.

Thank you for this suggestion. We have included a flow chart in the supplemental section to help outline the selection of patients.

The introduction talks mainly about RAL and LAM but these are not the usual malignant PEComas. It would have been interesting to develop more on these PEComas-NOS ( not otherwise specified) which are usually treated with mTOR inhibitors but often less subject to TSC1/2 mutation. In front of this very rare tumor, it would also be interesting to know if a double anatomopathological reading was performed by an expert

Thank you for this point. We included more types of PEComa in the listing in line 37. No double anatomopathological reading was performed. We took the pathologic data from the initial report and included it in the data.  These cases had all been read by pathologists with expertise in bone/soft tissue tumors.

INTRODUCTION

35 : there are also hepatic angiomyolipoma

Thank you for the suggestion; we have added this to the list in line 37

36  : PEComas of the lung are usually LAM or clear cell “sugar” tumor of lung, I don't understand the sub-type

Thank you for this suggestion; we have removed “PEComa of lung” and added “clear cell “sugar” tumor of lung.”

37 : Median survival was calculated on patients in the literature and not on the US population

Thank you for this suggestion; we have updated sentence to state that mOS is for patients in the literature and not exclusively in the US population.

38 : 70% in the literature

We have updated this value on line 40.

Materials and Methods

86 : RedCap database involves centralized review of PEComas cases by a qualified pathologist ?

Cases were reviewed by pathologists with expertise in bone and soft tissue tumors at time of specimen collection and diagnosis but there was no centralized re-review by a pathologist for the redcap data entry.

RESULTS 

Add a flow chart ?

Thank you for this suggestion; we have added a flow chart to the supplemental section.

Figure 1 : little interest due to the low number of TSC1 mutated patients

Thank you for the feedback. We agree with you but kept this table in case readers wanted to see a comparison between TSC-1 or TSC-2.

Figure 2 : 11 other mutations,  excluding tumors with multiple mutations ?

For this analysis, we included tumors with multiple mutations. Excluding them would lead to lower sample size even for the TSC-1 and TSC-2 mutations.

An other figure to see OS in mTOR-treated patients who are TSC mutated and non-mutated ?

Thank you for this suggestion. We would like to report this in a future publication; however, we would not be able to obtain this data within the five days given for resubmission of this publication.

Table 1 : angiomylipoma are classical triphasic or épithelioid angiomyolipoma ?

Angiomyolipoma were classic triphasic unless otherwise specified.

What is high or low grade ? It depend on Folpe's criteria ?

Thank you for clarifying this point. We included the number and percent of malignant pecoma based on folpe criteria in table 1 and stated Folpe’s criteria for malignant PEComa in lines 88-90 and cited the publication.

Why surgical resection end local recurrence are in "Grade" ?

We bolded these to indicate separate sections.

Table 2 : how we can have CBR without ORR for the population without mutation with other treatments ?

Thank you for clarifying this point. “Other” means that they have a mutation other than TSC-1 or -2. ORR was defined as PR+CR and CBR was defined as SD+PR+CR. The 3 patients had stable disease. This is why ORR is 0 but CBR is 66.7%. We included those definitions in the methods section.

DISCUSSION

Role of TFE3 translocation in the literature ? Response to mTOR in this group of PEComas ?

Thank you for this suggestion, We included a statement in the discussion and referenced an article regarding TFE3 translocation being associated with poor response to mTOR inhibition.

References

there are two references 1.

Thank you for pointing this out, we have resolved the issue.
